# Molecular Biology of Pediatric and Adult Ovarian Germ Cell Tumors: A Review

**DOI:** 10.3390/cancers15112990

**Published:** 2023-05-30

**Authors:** Mariana Tomazini Pinto, Gisele Eiras Martins, Ana Glenda Santarosa Vieira, Janaina Mello Soares Galvão, Cristiano de Pádua Souza, Carla Renata Pacheco Donato Macedo, Luiz Fernando Lopes

**Affiliations:** 1Molecular Oncology Research Center, Barretos Cancer Hospital, Barretos 14784400, Brazil; 2Brazilian Childhood Germ Cell Tumor Study Group, The Brazilian Pediatric Oncology Society (SOBOPE), Barretos 14784400, Brazil; 3Children’s Cancer Hospital from Hospital de Amor, Barretos 14784400, Brazil; 4Medical Oncology Department, Barretos Cancer Hospital, Barretos 14784400, Brazil; 5Pediatric Oncology Department, IOP/GRAACC/Federal University of Sao Paulo, Sao Paulo 04038001, Brazil

**Keywords:** ovarian cancer, germ cell tumors, genomics, epigenomics, pediatric and adult

## Abstract

**Simple Summary:**

Because ovarian germ cell tumors (OGCTs) are rare tumors, our current understanding of them is sparse; this is because few studies have investigated the molecular basis of pediatric and adult cancers. In this paper, we aim to provide an overview of the etiopathogenesis of OGCTs in children and adults, and we address the molecular landscape of these tumors.

**Abstract:**

Ovarian germ cell tumors (OGCTs) are rare in adults; indeed, they occur predominantly in children, adolescents, and young adults, and they account for approximately 11% of cancer diagnoses in these groups. Because OGCTs are rare tumors, our current understanding of them is sparse; this is because few studies have investigated the molecular basis of pediatric and adult cancers. Here, we review the etiopathogenesis of OGCTs in children and adults, and we address the molecular landscape of these tumors, including integrated genomic analysis, microRNAs, DNA methylation, the molecular implications of treatment resistance, and the development of in vitro and in vivo models. An elucidation of potential molecular alterations may provide a novel field for understanding the pathogenesis, tumorigenesis, diagnostic markers, and genetic peculiarity of the rarity and complexity of OGCTs.

## 1. Introduction

Germ cell tumors (GCTs) comprise a heterogeneous group of benign and malignant neoplasms. They are either located in the gonads (ovary or testis) or they are extragonadal. Clinically, histopathologically, genetically, and molecularly, these tumors vary significantly [1].

Germ cell & gonadal tumors are rare in adults; indeed, they occur predominantly in children, adolescents, and young adults. Moreover, they account for approximately 10% of cancer diagnoses in 15–19 years old group [2]. Ovarian germ cell tumors (OGCTs) are usually detected at an early stage, and they are associated with a favorable prognosis; however, the advanced stages of the disease, and cases where the patient relapses, show the long-term toxicity of common platinum-based regimens [3,4].

In recent decades, major efforts have been made to find new therapeutic alternatives to the platinum-based treatments, to understand the biology of GCTs, and to understand the molecular mechanisms of GCTs. Several studies have been performed in order to provide novel insights into integrated genomic analysis, microRNAs, DNA methylation, the molecular implications of treatment resistance, and the development of in vitro and in vivo models. The malignant germ cells that are found in the ovary often present numerous chromosomal variations and genetic abnormalities that are unrelated to the clinical findings. The molecular characterization of patients’ tumors may lead to the identification of prognostic and biomarkers.

This review describes the etiopathogenesis of ovarian germ cell tumors in children and adults, and it addresses the molecular landscape of these tumors. The rarity and complexity of OGCTs warrant collaborations between researchers and powerful clinical trial designs which include high-performance molecular characterization; this should be at the forefront of future research efforts.

## 2. Etiopathogenesis of Ovarian Germ Cell Tumors

Neoplasms arising in the ovary originate from different cell types which constitute the tissue of the ovary. The surface epithelium, the stroma, and the cellular elements of the follicle may give rise to distinct tumors; in particular, the cellular elements of the follicle can result in sex cord-stromal tumors or germ cell tumors [5]. OGCTs occur due to the pathologic transformation of the primordial germ cell (PGC) during the distinct stages of embryonic development (Figure 1).

The etiopathogenesis of OGCTs remains poorly understood; the cellular and molecular mechanisms of testicular germ cell tumors (TGCTs) have been studied to a greater extent. Both types of germ cell tumors are markedly aneuploid [6,7], and DNA methylation aberrations, the number of copied abnormalities, and mutations can contribute to the emergence of GCTs [8].

GCTs are a diverse group which is presumed to have a common cell of origin—the primordial germ cell (PGC). These tumors emerge in various midline or near-midline sites, and they are thought to be due to the arrested or aberrant migration of PGCs during embryogenesis. Moreover, the incidence and distribution of GCTs proves the importance of regulated cell death in midline sites [9].

### 2.1. Gonadal Dysgenesis

Disorders of sex development (DSD) comprise a series of diverse pathologies that are characterized by congenital conditions in which the development of chromosomal, gonadal, or anatomical sex is abnormal, thus resulting in the malformation of the internal and/or external genital organs [10,11]. DSD patients with gonadal dysgenesis (GD) present different somatic and genetic features with the development of hypoplastic genitalia [12,13]. Patients with GD and genotype 46, XY are at an increased risk of developing malignant OGCTs, which often emerge from gonadoblastomas (GB—a rare tumor containing both germ cells and sex cord-stromal cells) [4,13,14,15]. In such cases, the occurrence of Y chromosomal material confers a 30–40% risk of GB, and moreover, 50% of gonadoblastomas are correlated with dysgerminomas [16,17]. In collaboration with a group of German researchers, we analyzed bilateral OGCTs in clinically inapparent patients for sex chromosomal aberrations. We found that Y-chromosomal DNA sequences were detected in six tumors, in 15 patients with bilateral OGCTs [13].

The emergence of gonadoblastomas appears to be related to the *TSPY* gene, which is located on the short arm of the Y chromosome; it is responsible for encoding testis-specific protein-Y (TSPY) [18,19]. High levels of *TSPY* support the survival and proliferation of immature PGC, which remains in an embryonic state, with an increased expression of *OCT3/4* [13,20].

Germ cells from patients with GD might escape cell death if there is a persistent expression of both *OCT3/4* and *TSPY*; this is because it may later give rise to clonal expansion and neoplastic formation. Hermus et al. reported that 17 out of 19 (89%) of gonadoblastoma cases (with or without present dysgerminoma) showed positive staining with regard to the TSPY protein in neoplastic cells [18].

### 2.2. General Features of Embryonic and Gonadal Development

An understanding of gonad development and the characteristics of germ cells at different stages of differentiation is fundamental for comprehending the origin and features of several types of GCTs, particularly OGCTs [21].

The integration of three main events leads to PGC specification. These events are as follows: repression of the somatic program, reacquisition of potential pluripotency, and genome-wide epigenetic reprogramming [5,22]. In the embryonic phase, PGCs develop in the yolk sacs of embryos, and they migrate to the gonads through the midline of the body in a process controlled by the stem cell growth factor receptor, KIT (KIT), and its ligand, KIT ligand (KITLG) [5,8]. Embryonic stem cells (ESCs), PGCs, and germ cells are proliferating cells containing active telomerase, and they are able to replicate indefinitely. In addition to potentially infinite replicative abilities, PGCs share other hallmark characteristics with cancer cells, such as anaerobic glycolysis and the ability to migrate [22,23]. A feature of such ESCs is the expression of OCT4, SOX2, and NANOG, which maintain pluripotency [24].

Runyan et al. showed that the process of targeted migration and controlled cell death are essential for the possible localization of germ cells in the genital ridges. The study showed that the regulation of apoptosis during migration causes the removal of midline germ cells and those pro-apoptotic genes of the intrinsic pathway, which are upregulated in migratory germ cells [25].

ESCs that are derived from the preimplantation inner cell mass (ICM) and epiblast, which present totipotent developmental potential (so-called naïve state), are distinguished by a permissive epigenetic signature, as well as two active X chromosomes (in female cells) [22,24]. The expression of OCT4 is driven by a distal enhancer, and OCT4 partners with SOX2 (SRY-box 2). Between human embryonic weeks three and four, these cells enter a primed state by submitting female X inactivation and promoting methylation in pluripotency genes, thus implementing a restricted self-renewal ability and pluripotent developmental potential. During the same period, PGCs are specified, and are subsequently the only OCT4-expressing cells in the embryo. During human PGC migration, approximately weeks five and six, OCT4 switches SOX2 for SOX17 (SRY-box 17). Moreover, SOX17 and BLIMP1 (B lymphocyte-induced maturation protein 1) are also important for maintaining the wellbeing of PGCs by preventing them from being reprogrammed to an ESC [22,24,26].

When PGCs are in a latent potency state, which is determined by their epigenetic status, they can be reprogrammed to enter a primed state. During GCT development, ESCs may enter a naïve state. In addition, all the GCT subtypes present the same global methylation and genomic imprinting patterns of the GCT stem cells, which strongly resemble those of their normal counterparts [22]. Imprinted genes are localized within various chromosomal clusters that contain GC-rich regions with differentially methylated CpG dinucleotides. Genomic imprinting patterns are responsible for allele-specific gene expression. Imprinted genes in mammals have specific roles to play in the developing germ cells. As opposed to to somatic cells, which maintain the parent-specific imprinting pattern, germ cells must, at some stage, erase the imprinted genes and establish a new, sex-specific imprinting template [27,28].

Once PGCs reach the bipotential gonads, the absence of the *SRY* gene, which is typically located on the Y chromosome or its downstream target *SOX9*, and the subsequent activity of female-specific genes, promote the development of the gonads into ovaries. A subset of ovarian-specific genes controls the ovarian morphogenesis process [21]. In fetal ovaries, at around the tenth week of gestation, the retinoic acid produced by the mesonephros stimulates the initiation of meiosis, which appears to occur asynchronously. Indeed, as increasing numbers of germ cells initiate meiosis, some oogonia still express stem cell markers and continue to proliferate until at least week 16 [5,21]. After the 12th week, the oogonia seem to segment into two cell populations: a KIT/OCT4 subset located at the periphery and a VASA positive subset that is located near the center of the ovary. A few genes control the mitotic-to-meiotic transition, and any dysfunction during this transitional period can be a root cause of the conversion of PGCs into GCTs [21].

Following gonadal colonization, proliferating oogonia arrange themselves into cyst-like structures with supporting pregranulosa cells by the end of the seventh month of gestation. Furthermore, at this point, they rapidly lose their POU domain, as well as their class 5 and transcription factor 1 (POU5F1) expression, and mitotic activity is terminated. At this stage, almost all oogonia enter meiotic prophase I and become primary oocytes, where they remain arrested, and they subsequently form primordial follicles [5,21]. Ovary differentiation occurs due to feminizing factors WNT4 and FOXL2; the latter factor has been termed the gatekeeper of ovarian identity [5,22].

### 2.3. OGCT Development

GCTs are rarely caused by somatic driver mutations; indeed, they are the result of reprogramming PGCs due to a failure in the cell process to control their latent developmental potency. This not only explains their developmental potential, but it also describes the diverse clinical and pathological aspects of GCTs [22].

In accordance with the model of tumorigenesis proposed by Teilum, germinomas (dysgerminomas in ovarian sites) emerge directly from primordial germ cells, and consequently, they retain their pluripotent state. Embryonal carcinomas (ECs) presenting early embryonic differentiation can give rise to tumors containing all three germ layers (endoderm, ectoderm, and mesoderm). In contrast, PGCs that exhibit extra-embryonic differentiation result in either yolk-sac tumors (YSTs) or choriocarcinomas (CC). The mixed germ cell tumors contain various malignant histologies [29,30]. Dysgerminoma and immature teratoma are the most common types of OGCT, comprising 65–70% of all OGCTs, followed by YSTs (14.5%), and finally, mixed GCTs (5.3%) [31,32].

Most OGCTs are unilateral; however, 10–15% of dysgerminoma and 5–10% of the mixed OGCT subtype may be bilateral [33,34].

Furthermore, PGCs expressing pluripotent genes, such as *NANOG* and *POU5F1*, and which express complete demethylation, gain additional genomic abnormalities, such as the KIT mutation, or the isochromosome 12p; subsequently, this causes the emergence of dysgerminomas (DGs). PGCs expressing *DPPA3* sometimes gain isochromosome 12p and they subsequently develop into ECs. PGCs expressing *DPPA3* restore DNA methylation and differentiate into sperm or egg cells. During differentiation, further genomic abnormalities are acquired by PGCs, which can develop into YSTs or teratomas. Teratomas, ECs, and YSTs imprint in a sex-specific manner [8].

Pediatric and adult GCTs differ from the imprinting patterns of loci in, for example, *IGF2/H19*; this means that pediatric GCTs tend to emerge from more immature PGCs. In addition, compared with germinomas, pediatric YSTs show increased methylation in several genes’ regulatory loci, and they demonstrate a methylator phenotype, such as a decrease in the number of genes that programme cell death and repress WNT signaling [27,30].

Kato et al. investigated genetic zygosity in a series of mature ovarian teratomas, struma ovarii, and ovarian carcinoids. They showed that homozygous genotypes were present in 50% of mature teratomas, 50% of struma ovarii, and 33% of ovarian carcinoids; this suggests that the oocyte that emerges after meiosis I is an important factor that causes these tumors [6].

Multi-region whole exome sequencing of immature ovarian teratomas (in the range of 8–29 years) was performed, and the results revealed that this type of tumor is characterized by 2N near-diploid genomes which have undergone a severe loss in heterozygosity. In addition, they exhibit an absence of the genes which harbor recurrent mutations or known oncogenic variants. Moreover, different patterns in the left and right ovaries displayed a loss in heterozygosity, thus suggesting that bilateral ovarian teratomas emerge independently of one another. Altogether, these outcomes show that numerous meiotic mistakes can form genetically distinct tumors; these tumors are unique as a result of their stark allelic imbalances, a lack of somatic mutations, and copy number alterations [35].

## 3. Search for Better Therapeutic Approaches

This section may be divided by subheadings. It should provide a concise and precise description of the experimental results, their interpretation, as well as the experimental conclusions that can be drawn. Historically, tumors have been removed via an open approach in order to preserve capsular integrity. In addition, conservative surgery to preserve fertility has been established as the standard of care [36]. The post-operative management of OGCTs has evolved over time. The role of ‘second look’ surgery has been investigated in several studies conducted by the Gynecologic Oncology Group (GOG). Moreover, it is not recommended after a complete clinical response to surgery and when postoperative chemotherapy has occurred. Nevertheless, second look surgery may be recommended for patients with residual masses after the initial treatment stage, particularly in cases that present teratomatous elements, or when tumor markers continue to be present [34,37,38].

A wide variety of chemotherapy regimens have been used for OGCTs, and there are some differences and similarities in the management of OGCTs in pediatric and adult patients. For adult OGCT patients with dysgerminomas and grade I IT, postoperative observation is recommended; however, postoperative chemotherapy is advised for all other histologies. Conversely, observation only is recommended for pediatric patients, with any stage-I-tumor, with any histology [34].

CDDP-based chemotherapy is the first line of chemotherapy treatment, and it can cure most patients with GCTs [39]. The platinum-based regimen is recommended for post-operative chemotherapy in both pediatric and adult patients, wherein bleomycin, etoposide, and cisplatin (BEP) is usually the offered standard of care.

In order to better manage GCT pediatric patients, the Brazilian Childhood Germ Cell Tumor Study Group devised the first national protocol with regard to these tumors in 1991 (GCT-91 protocol) (Figure 2). The main goal of this protocol was to assess a standard risk-adapted and response-based treatment approach for pediatric GCTs in Brazil; this approach featured a two-drug regimen comprising PE (Platin and Etoposide). The GCT-91 protocol included 106 GCT pediatric patients (in the range of 0–18 years). Of these patients, 71 were treated with chemotherapy in addition to surgery. As expected, the majority of patients (*n* = 53, 75%) who received chemotherapy had advanced stage disease (stage III to IV). For those who received intensified chemotherapy with two drugs (*n* = 22; eight testis tumors and fourteen ovarian tumors), the overall survival (OS) rate was 86.4%. Conversely, for the 11 patients (four testis tumors and seven ovarian tumors) who were treated with five drugs, the 5-year OS rate was 54.5%. Our data suggested that the treatment with the complex of three agents (BEP) may not be required to achieve long-term survival, including in patients who had advanced stage disease [40].

Regarding the subsequent protocol (GCT-99), we expanded the scope of this investigation into the standard risk-adapted and response-based treatment approach by adapting the number and types of chemotherapy cycles for patients who were considered to be at intermediate risk (IR) or high risk (HR) on the basis of their response to chemotherapy treatment. The second Brazilian protocol, GCT-99, included 480 GCT pediatric patients (in the range of 0–18 years), of which, 206 had ovarian tumors; 80 were low risk (LO), 97 were IR, and 29 were HR. We showed that the treatment with two drugs did not compromise survival outcomes for IR patients that exhibited a good response to chemotherapy, and therapy with PEI did not significantly improve OS and event-free survival (EFS) rates in HR patients. In addition, the 10 year EFS for patients in the IR group and HR group were higher in ovarian cases compared with tumors originating in the testes and other sites [41].

In 2009, the Malignant Germ Cell International Consortium (MaGIC) was launched to advance the search for a cure for GCTs. Using MaGIC data commons, a risk assessment to stratify malignant extracranial pediatric GCTs was performed. In the multivariable analysis, in children aged 11 years or older, the tumor site (categorized as testicular v ovarian v extragonadal) and a stage IV classification were significant factors that were associated with poor outcomes. Moreover, the analysis showed that a group of patients aged 11 years or older, with stage IV ovarian tumors, had predicted long-term disease-free (LTDF) survival rates of less than 70% [42].

In adult OGCT patients with dysgerminoma IA or IB, or with immature teratoma G1, the guidelines recommend observation [43]. For women that have dysgerminoma at a stage that is more advanced than IB, immature teratoma at stage IA G2–IV, or other non-dysgerminomas, endodermal sinus tumors, embryonal carcinoma, or choriocarcinoma in stages I to IV, adjuvant chemotherapy is recommended [44,45,46]. The guidelines recommend bleomycin, at 30 UI IV, as a bolus, on D1, D8, and D15, plus cisplatin, 20 mg/m^2^ IV, for 30 min, on D1 to D5, and etoposide, 100 mg/m^2^ IV, on D1 to D5 (BEP), repeated every three weeks, for a total of three cycles if optimal debulking occurs, or four cycles if suboptimal debulking occurs [47,48]. For patients with dysgerminoma stage IB-III, or nonseminomatous tumors with some contraindication to cisplatin and/or bleomycin, an alternative option would be to use carboplatin, 400 mg/m^2^, on D1, and etoposide, 120 mg/m^2^, on D1 to D3 IV, every four weeks, for three cycles [49,50]. In patients who have unresectable high-volume disease, first, neoadjuvant chemotherapy with BEP may be performed, for four cycles, followed by cytoreduction, after which, surgery may be possible [51].

Adult OGCTs patients who present progression after the first line of chemotherapy has been performed are usually treated with ifosfamide and CDDP-based regimens. This approach is also used in men with relapsed GCTs. For patients with residual disease, and for patients with relapsed GCTs, the second-line therapy comprises paclitaxel, ifosfamide, and CDDP (TIP) [39,52] (Figure 2).

In the years since this protocol was first published, this risk stratification has been validated in three independent datasets, as provided by our Brazilian research group, together with the British and French clinical trial groups. The original MaGIC risk stratification method was confirmed, thus supporting its use in prospective clinical trial designs [53].

Since adult and pediatric medical practices are noticeably different, Newton et al. evaluated whether reduced toxicity treatment could be extended to patients that are older than 18 [54]. To achieve this, a multicenter cohort study (138 pediatric and adult OGCT patients) was carried out in four large UK cancer centers over 12 years. These treatments, which were less toxic than chemotherapy, showed an excellent survival rate in contrast with chemotherapy; indeed, chemotherapy promotes a high level of toxicity in the body, and the positive outcome of this study suggests that chemotherapeutic methods constitute significant overtreatment. This observation supports the idea of extending reduced toxicity, pediatric regimens to adults [54].
Figure 2Timeline of treatment evolution in adult and pediatric patients with OGCTs. The description of progress in pediatric treatment was based on the Brazilian experience and the Malignant Germ Cell International Consortium. PE: Platin and Etoposide. NSGCTs: Nonseminomatous. TIP: Paclitaxel, Ifosfamide, and CDDP [39,40,41,42,47,48,49,50,53,54].
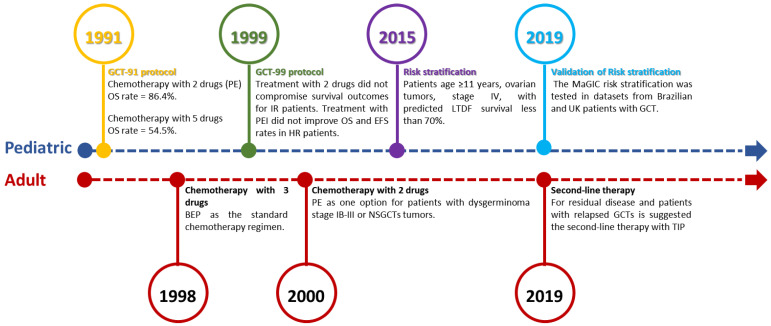


Although international guidelines still recommend BEP in nearly all GCT cases, studies have shown that this promotes widespread overtreatment, and thus, a reduction in BEP use is necessary. Our group has produced a satisfactory result with lowered cisplatin dosages and no bleomycin in the treatment of pediatric GCT patients [41]; however, further studies should be devoted to determining which drugs can be used to avoid overtreatment over a certain period of time. To address this issue, we pose the following question: What is the best adjuvant chemotherapy regimen for patients who require chemotherapy? Are second line treatments necessary for patients with recurrent OGCTs? Is it possible to extend reduced toxicity pediatric regimens to adults? Are treatment-induced toxicities reduced by using fewer drugs/agents? The answer to these questions will require ongoing international collaborations between pediatric and adult medical oncologists, and prospective clinical trials involving these patients will need to be performed.

## 4. Molecular Biology

As ovarian GCTs are rare, there is a lack of research on this subject. Moreover, only a few genetic studies have been performed on malignant GCTs, on patients in the pediatric age group. More extensive studies are required to understand the genetic variations between these tumors. Below, we highlight the main evidence showing the genetic, cytogenetic, and epigenetic alterations in GCTs. Recently, our group has published a review that highlights the molecular biology of pediatric and adult male GCTs [55]. Moreover, in this review, our aim is similar, but we intend to emphasize female GCTs (Figure 3).

### 4.1. Genetics and Citogenetics

In general, pediatric cancers are known to have low mutational burdens, or ‘quiet’ genomes, relative to adult tumors; indeed, pediatric malignant tumors have approximately 1000-fold fewer somatic mutations than several adult cancers [56]. Previous studies have shown that pediatric and adult GCTs differ biologically in several aspects [57,58,59], but most of these studies have been performed on testicular GCTs [55], and little is known of the differences between ovarian GCTs.

The cytogenetic features in GCTs differ according to age and histological subtype [60]. Practically, both pediatric and adult teratomas have a normal karyotype, and this feature in ovarian teratomas suggests that they arise from an oocyte after meiosis [6,60].

Genetic changes associated with malignant OGCT development have been investigated over the years. The gains and losses obtained via comparative genomic hybridization were analyzed in 21 malignant OGCTs (in the range of 2–28 years) and the most common regions that made gains included chromosomes 12p, 21, 8, and Iq. Conversely, the most common regions that suffered losses included chromosome 13. Patients younger than 15 years old made gains on 12p (10 of 14 tumors), and half of these gains occurred due to dysgerminomas [61]. In accordance with these results, Kraggerud et al. analyzed a series of archival OGCTs, including five pediatric patients (in the range of 11–15 years; mean age, 12.4 years) and 20 adult patients (in the range of 16–70 years; mean age, 29.5 years) [62]. The highest average number of changes was found in patients with dysgerminomas, followed by patients with yolk-sac tumors, and finally in patients with immature teratomas. The most common changes in dysgerminomas were due to the gains made by chromosome arms 1p, 6p, 12p, 12q, 15q, 20q, 21q, and 22q; gains made by the chromosomes 7, 8, 17, and 19 in their entirety; and losses from 13q. Gains made by the chromosome arms of 12p occur more frequently in patients older than 15 years (9 of 11 tumors). Altogether, these results suggest that ovarian dysgerminomas and yolk-sac tumors share the same genetic pathways as testicular GCTs; however, because immature teratomas did not exhibit gains made by 12p, and as they typically showed fewer changes, we may assume that they emerge as a result of distinct pathogenetic mechanisms [62]. In addition, the data from the two studies described above showed that patients younger than 15 years, and patients older than 15 years, make similar 12p gains; these gains also occur at a similar rate. It is worth mentioning that the most commonly observed change in all histological subtypes of adult testicular GCTs occurs in the short arm of chromosome 12; this is less frequent in pediatric patients, thus indicating that TGCTs and OGCTs evolve through some of the same pathogenetic mechanisms in both sexes [63,64].

Subsequently, Palmer et al. used metaphase-based comparative genomic hybridization to analyze 34 malignant pediatric GCTs (in the range of 0–16 years), including 17 OGCTs [9]. Of these 17 OGCTs, one occurred in a child younger than 5 years old at diagnosis, and 16 occurred in older children. YSTs showed significant gains on 3p, and significant losses on 1p and 6q as compared with germinoma, with an increased number of losses on 4q. YSTs in children older than 5 years (typically ovarian) showed fewer imbalances as compared with the YSTs of children younger than 5 years. Moreover, germinoma showed a greater number of gains on 12q and 19q, and more losses on 11q. Gains made by chromosome 12p was observed in 44% (15 of 34 cases) of malignant GCTs, and such gains were more frequent if the patient was suffering from a germinoma tumor. Four of the fourteen YSTs in children younger than 5 years old showed 12p gains. The regions where gains were made are as follows: chromosome 12, 12p11-pter, 12p12-pter, and 12p13-pter [9].

A comprehensive genomic analysis of pediatric GCTs has been performed using 51 pediatric GCT samples (in the range of 2 months to 19 years); of these samples, 16 tumors developed in the ovaries. Global uniparental disomies (UPDs) were detected in the gonadal samples of female YSTs; here, the ovarian YST samples showed global UPDs that were similar to ovarian teratomas. Considering all samples (*n* = 51), the gains made by the long arms of chromosome 20 comprised the most frequently detected copied alterations (57%), followed by the gains made by the short arms of chromosome 12 (39%), of which, 40% carried isochromosome 12p; the ages of the children in this latter group was >10 years. The same group of children (>10 years) studied in the YST samples exhibited *KIT*, *KRAS*, and *NRAS* mutations [8]. Whole-exome and RNA sequencing was performed in YST patients (*n* = 30), and other somatic driver candidates were identified, including the significantly mutated genes, *KRAS* and *KIT*, and drivers of copied alterations, including the deleted *ARID1A* and *PARK2*, and the amplified *ZNF217*, *CDKN1B*, and *KRAS* [65].

In accordance with the AACR Project GENIE Consortium data (Available online: www.mycancergenome.org (accessed on 15 June 2021).) [66], ovarian germ cell tumors most frequently harbor alterations in *TP53*, *KRAS*, *KMT2D*, *PIK3CD*, and *PIK3CA*. Indeed, previous studies reinforce the occurrence of genomic alterations in *KIT*, *KRAS*, and *PIK3CA*, as well as mutations and the number of copied alterations in these genes [4,5,7,8,65,67]. *KIT* mutations in OGCT subtypes, such as dysgerminomas, gonadoblastomas, and yolk sac tumors, are most frequently exhibited in exon 17; these mutations can lead to increased chances of survival and the proliferation of undifferentiated oogonia [5].

A large genetic landscape study of 87 OGCTs (in the range of 4–47 years) reported that the overall mutation rate in OGCTs is very low compared with other tumors; the mutation rate is the same as in dysgerminomas and non-dysgerminomas, and the mutation rate for patients aged <18 years and patients >18 years is identical. It was also shown that recurrent mutations in *KIT*, and gains made by the chromosome 12p, are considered to be the keystone features of OGCTs [7], and they are invariably found in adult MGCTs as well [68]. *KIT* was the most significantly mutated gene, with five non-synonymous mutations in four samples (4/24–16.7%). The *KRAS* mutation was also recurrent, but mutations occurred less frequently than with *KIT*, with two *KRAS* mutations in codon 12 (one in embryonal carcinoma and one in mixed yolk sac-dysgerminoma tumors). Moreover, the analysis also revealed several known oncogenes with plausible driver roles in *FIP1L1*, *BUB1B*, *CASC5*, and *AKT1*; no p53 mutations were observed [5,7]. *PIK3CA* gene amplification was observed in 21.8% (19/87) of cases and *AKT* was amplified in 20.6% (18/87) of cases, thus suggesting an enrichment of the *PI3K* pathway and the use of PI3K-inhibitors in recurrent OGCTs [7].

The histological subtype of the yolk sac tumor is the second most common malignant OGCT, and some of these tumors have specific molecular features. For example, YSTs show a distinct overexpression of genes that are related to the WNT/β-catenin and TGF-β/BMP pathways as compared with dysgerminoma and GCTs [69,70]; however, YSTs are among the cancer types that are characterized as ‘unwell’ at the molecular level [65]. Through the whole–exome sequencing of the YST tumor sample set, significantly mutated genes, including mutations of the *KRAS* and *KIT* genes, and infrequent *TP53* mutations, were observed. In addition, the number of copied alteration drivers, including the deleted *ARID1A* and *PARK2* genes, and the amplified *ZNF217*, *CDKN1B*, and *KRAS* genes, were also detected in YSTs. Moreover, *OVOL2* overexpression was associated with YST resistance to cisplatin [65].

Although adult and pediatric GCT patients share some similarities that are related to certain molecular changes, they also share some differences. For example, in adults, the *BRAF* mutations were related to microsatellite instability and deficient mismatch repair protein expression [71]; conversely, in pediatric GCT patients, neither *BRAF* nor *KRAS* mutations, nor losses caused by *MLH1* and *MSH6* (markers for microsatellite instability) expression, were found [57].

Overall, the mutation rate in OGCTs is very low compared with other tumors. After analyzing all the information concerning comparative genomic hybridization, it is possible to argue that 12p gains are more frequently found in adult TGCTs and OGCTs, and they are less frequently found in childhood TGCTs and OGCTs. Moreover, gains made by 12p11-12 are common in adult OGCTs, whereas gains made by 12p13-pter are more frequent in childhood. As testicular GCTs tend to recur more frequently, new insights and data from clinical trials that focus on such GCTs can be applied to the treatment of OGCTs, and thus increase the therapeutic options for this rare disease. In addition, molecular alterations in genes *AKT1*, *ARID1A*, *BUB1B*, *CASC5*, *CDKN1B*, *FIP1L1*, *PIK3CA*, *KIT*, *KMT2D*, *KRAS*, *NRAS*, *OVOL2*, *PARK2*, *PIK3CD*, *TP53*, and *ZNF217* have also been found in adult and pediatric OGCTs (Figure 3); however, there are some discrepancies between studies. Therefore, further analysis of whole exome sequencing is required to detect gene mutations that are associated with pediatric and adult patients with OGCTs.

### 4.2. Epigenetics

Epigenetic mechanisms have been intensely studied in recent years. The study of epigenetics involves an examination of the key processes of DNA methylation, chromatin modifications, nucleosome positioning, and alterations in noncoding RNA profiles; therefore, these processes regulate several aspects of one’s biology that are essential to the genesis of cancer [72]. In this section, we summarize and discuss the published studies on epigenetics that have a specific focus on OGCTs.

#### 4.2.1. DNA Methylation

DNA methylation is a covalent modification of the 5-carbon on cytosine residues (5mC) in CpG dinucleotides. Alterations in DNA methylation were some of the first alterations to be described with regard to this subject, and these alterations are some of the most explored in terms of epigenetic modifications in cancer. These alterations include hypermethylation, hypomethylation, and loss of imprinting (LOI) [73,74].

DNA methylation is an epigenetic marker that is associated with gene silencing. Convention stipulates that DNA methylation functions predominantly for irreversible silence transcription; however, this concept is being challenged. Methylation is crucial for the development of an organism, as it controls gene expression, chromosomal integrity, and recombination events. It is known that gene silencing, via epigenetic events, does not always cause problems; however, there are some factors that may cause the methylation process to become abnormal, such as local alterations in DNA structure, exposition to carcinogens, and an increase in methyl transferase DNA activity [75]. Methylation levels are significantly altered when the cell becomes malignant, and this change may result from the imbalance between hypomethylation and hypermethylation. Imbalanced hypermethylation promotes gene instability, which leads to cell proliferation, thus causing the loss of tumor suppressor gene function. On the other hand, hypomethylation is related to tumor progression; this is because it activates proto-oncogenes [76].

DNA methylation is characteristic of many types of cancer, and it may be the key factor in the production of germ cell tumors due to the extensive epigenetic reprogramming that occurs in the germ line during normal development [77]. Genes that are differentially methylated may provide insights into GCT etiology; this may even include the timing of GCT initiation [78]. In the future, methylation may help stratify treatments.

Global DNA methylation was investigated in GCTs (*n* = 251), and the results showed that undifferentiated GCTs, including seminomas, unclassified intratubular germ cell neoplasia, and gonadoblastomas, are hypomethylated, whereas GCTs that are differentiated to a greater extent (teratomas, yolk sac tumors, and choriocarcinomas) are hypermethylated. In addition, embryonal carcinomas exhibited intermediate patterns [79].

The major studies that concern methylation in GCT have been performed in testicular tumors, and thus, little is known of methylation processes in OGCTs. Yu et al. compared the DNA methylation processes of embryonal carcinoma and seminoma in TGCTs using datasets concerning mRNA expression and DNA methylation profiling. A total of 37 genes were presented, thus providing data on both mRNA expression and DNA methylation changes. Moreover, five of them (*PRDM1*, *LMO2*, *FAM53B*, *HCN4*, and *FAM124B*) were downregulated and showed a high degree of methylation in embryonal carcinoma; these genes were significantly associated with relapse-free survival [80].

The hypermethylation signature was performed in male GCTs by analyzing 21 gene promoters. Nonseminomatous tumors (NSGCTs) showed a 60% methylation rate in one or more gene promoters, whereas seminomatous tumors exhibited almost no methylation. In addition, *MGMT*, *RASSF1A*, *BRCA1*, and *HIC1* were frequently methylated in NSGCTs [81].

Insulin-like growth factor 2 (IGF2) and H19 are a pair of mutually imprinted genes [82], and their expression is related to the methylation status of the *IGF2/H19* imprinting control region (ICR). An analysis of the *IGF2/H19* imprinting control region was performed in 55 GCT patients (24 adults and 31 children/adolescents). Most GCT patients exhibited a low degree of methylation in the *IGF2/H19* ICR regions, and all 8 ovarian GCTs were hypomethylated, thus suggesting that the methylation analyses of ICR offer a foundational understanding of the early stages of GCT formation [83].

DNA methylation array analysis was performed in 51 GCT pediatric patients, including 6 germinomas, 2 embryonal carcinomas, 4 immature teratomas, 3 mature teratomas, 30 yolk sac tumors, and 6 mixed germ cell tumors [8]. Germinomas exhibited global hypomethylation and the upregulation of pluripotent genes; moreover, those genes were also overexpressed in ECs. YSTs exhibit an overexpression of endodermal genes, including *GATA6* and *FOXA2*, which were hypomethylated. Moreover, DNA methylation patterns were shown to be different in infants who were suffering from YSTs and in older children suffering from YSTs; therefore, each GCT subtype possessed unique characteristics [8]. In accordance with these results, the Children’s Oncology Group evaluated differences between DNA methylation profiles in 154 pediatric GCTs [78]. Tumor specimens included 54 mixed types, 9 teratomas, 70 YSTs, and 21 germinomas/seminoma/dysgerminomas, of which, 58% were found in the ovary (dysgerminoma). Moreover, 8481 differentially methylated regions (DMRs) were identified, and it was reported that germinomas exhibited lower levels of methylation as compared with the other histologic subtypes of GCTs. Pathway analysis revealed that germinomas/seminomas/dysgerminomas exhibited a decreased level of methylation during angiogenesis and in immune cell-related pathways, as compared with YSTs. Finally, YST exhibited hypermethylation in tumor suppressor genes, thus suggesting a prospective mechanism for tumor initiation and resistance to chemotherapy [78].

As previously discussed, there is a distinct methylation pattern that occurs in accordance with GCT histologies, including in OGCTs. Amatruda et al. analyzed the differences between DNA methylation processes in 51 pediatric and adolescent GCT patients (in the range of 0–21 years), including 24 patients with tumors in the ovary, 7 with tumors in the testis, and 20 patients whose tumors were extragonadal; furthermore, these tumors were classified into different histological types (YSTs, germinomas, and teratomas). Ovarian teratomas exhibited reduced methylation levels in the loci that are typically methylated on the paternal allele. Furthermore, increased methylation levels were found in the loci that are typically methylated on the maternal allele. Methylation that occurs on imprinted loci causes the differentiation between extragonadal locations and ovarian teratomas, the latter of which exhibits hypomethylation and hypermethylation on the CpG loci, which typically methylates on the paternal allele and maternal allele, respectively [77]. These data emphasize the idea that the methylation status of the imprinted loci in GCTs characterizes the origin and stage of development of the PGC when the transformation occurred [59].

It can be assumed that there are differences in methylation pattern with regard to the main histologic subtypes of pediatric GCTs across all sites; moreover, methylation differences occur across the different pediatric age groups (Figure 3). Further multi center studies are needed so that a larger sample can be examined, and so that these patterns can be more accurately defined. As a result, in the near future, studies concerning treatment for hypo methylating drugs may become a reality, not only for adult testicular tumors and adult ovarian tumors, but also for pediatric GCTs.

#### 4.2.2. MicroRNA

MicroRNAs (miRNAs) are small, endogenous, non-coding RNAs (noncoding RNA) that are phylogenetically conserved and formed, on average, by a sequence of 18 to 25 nucleotides [84]; these nucleotides play an important role in post-transcriptional regulation, via cleavage or the translational repression of the target messenger RNA (mRNA).

Several miRNAs regulate diverse functions in cells, such as proliferation, survival, and apoptosis. Moreover, their aberrant expression has been closely associated with many diseases, including cancer [85,86,87]. MicroRNAs affect the development and progression of cancer cells by binding to RNAs and regulating their expressions [88].

One of the first studies that evaluated miRNAs in testicular GCTs also performed functional genetic screening in order to identify miRNAs that cooperate with oncogenes during cellular transformation. Two miRNAs were identified, miR-372 and miR-373, which each permitted proliferation and the tumorigenesis of human cells that contain oncogenic RAS and the active wild-type p53 [89].

In the pediatric GCTs, Palmer et al. investigated the global miRNA profiles that address gonadal and extragonadal sites. They analyzed the variations between adult malignant gonadal GCTs. The most significant overexpressed miRNAs in malignant GCTs were found in miR-371~373 and miR-302 clusters, which showed an increase in expression regardless of histological subtype, site, or patient age. These data suggest that the 371~373 and miR-302 clusters play a central role in the pathogenesis of GCTs by downregulating target genes [90].

The evolution of research in this field has led to the exploration of the role of miRNAs in the serum of patients with a GCT, which has subsequently been compared with healthy or sick individuals. Once again, testicular disease is the focus of a study by Syring et al. [91], which demonstrates that serum levels of miR-367-3p, miR-371a-3p, miR-372-3p, and miR-373-3p were significantly increased in patients with TCGTs as compared with healthy individuals. In particular, miR-371a-3p exhibited sensitivity and specificity levels of 84.7% and 99%, respectively, thus outperforming the human chorionic gonadotropin or alpha-fetoprotein test. Even the serum level of this miRNA decreased after the localized tumor was resected, thus indicating a targeted release of the serum by the tumor. Dieckmann et al. argued that as a result of confirmed data in a multicenter prospective study, involving more than 600 serum samples from patients with TCGTs (range, 16–69 years) [92], and in another study that used serum from a 4 year old male pediatric patient, miRNAs may therefore be considered as biomarkers of GCTs [93].

Malignant and benign OGCTs were characterized in accordance with the miRNA profiles of 16 adult patients (in the range of 17–73 years). For malignant tumors, the miR-548 family, miR-302, and miR-371~373 clusters were overexpressed, whereas let-7 family members were downregulated compared with the benign OGCTs. For benign tumors, miR-193b-5p/3p, miR-320a/b, and miR-22-5p levels were frequently higher than those expressed by the malignant OGCTs [94]. These data suggest that miRNAs may be used as a potential tool for defining histological subtypes and biological differences in adult OGCTs (Figure 3).

In addition, miRNA expression was also associated with cisplatin-resistant GCT cell lines; in turn, this was associated with the upregulation of miR-512-3p, miR-515, miR-517, miR-518, and miR-525, and the downregulation of miR -99a, miR-100, and miR-145, as well as a cisplatin-resistant phenotype in human GCTs [95]. Further functional studies are required to gain an awareness of the role of miRNAs in drug resistance.

Integrated molecular profiles comprising dysgerminomas, seminomas, and YSTs were reported in both sexes. By comparing the global miRNA profiles of the histological subtypes, it was found that dysgerminomas and seminomas cluster separately from the YSTs of both male and female patients. The upregulation of the mir-302–367 and mir-371–373 clusters was the most significant finding, as it occurred regardless of histological subtype, tumor site (ovary, testis, or extragonadal), and patient age [90,96,97,98].

Due to the low incidence of malignant OGCTs, the sample size precludes exclusive molecular studies of unicentric ovarian GCTs; however, in the transcriptome profile, the mir-371–373 and mir-302–367 clusters in OGCTs were suggested as potential serum biomarkers; this is because the specific miRNA expression pattern in each of the histological subtypes of these ovarian tumors are clear [5].

Future miRNA studies will reveal novel information on the role of this molecule in the development of OGCTs in adult and pediatric patients, and they will attest to the potential value of miRNAs as tumor markers.

## 5. In Vitro and In Vivo Models

Immortalized cancer cell lines are derived from patient tumors, and they are manipulated and maintained in vitro to proliferate indefinitely. These cell lines are the most frequently used experimental model in cancer research because they preserve numerous tumor properties, and they have been of immense value with regard to the comprehension of cancer biology and in the development of new therapeutic approaches [99]. Nevertheless, there are important differences between the molecular and genetic profiles of cell lines and tumors. Although cell lines share many features of tumors, they obtain additional alterations during the immortalization process, and during cell growth and maintenance when they are cultured [100].

Due to the absence of models for ovarian GCTs, it is difficult to identify a target for a novel molecular-based therapy, and thus, it is difficult to improve on therapeutic strategies in cases of relapse. Based on this issue, Shibata et al. (2008) established and characterized a human ovarian YST cell line for the first time [101]. The cell line was taken from a 28 year old woman who underwent a right salpingo-oophorectomy to treat ovarian YST cell lines (NOY1 and NOY2) (Figure 3). In subsequent studies, the same research group established a model that exhibited cisplatin-resistant properties using NOY-1 cells (NOY1-CR) in order to investigate the mechanism driving cisplatin resistance [102]. NOY1-CR was cultured using stepwise exposure methods (from 0.5 µg/mL cisplatin) for 12 months, and this cell line became 22.3 times more cisplatin-resistant than its parent cells. In order to identify the genes associated with cisplatin resistance in NOY1-CR cells, cDNA microarray analysis was performed. Data showed that the *GSTA1* gene was overexpressed in NOY1-CR, and the inhibition of *GSTA1* restored cisplatin sensitivity, thus suggesting the potential for *GSTA1* to become a novel therapeutic target for cisplatin-resistant ovarian YSTs [102].

Several molecular analyses were performed using NOY1-CR [103]. Copy number analysis was conducted using the methylation intensity data, which showed losses on chr7p, chr15q, chr16q, and chrX, and gains on chr3p and chr13qtel. The most methylated genes/promotors exhibited reduced methylation levels in NOY1-CR. Gene expression was evaluated, and reduced levels of gene and promoter methylation in the resistant cells correlated with the increased expression of *ALDH3A1* and *RP11-311F12.1* genes. Resistant cells exhibited an increased expression of prominin-1 (*CD133*), ATP binding cassette subfamily G member 2 (*ABCG2*), and aldehyde dehydrogenase 3 isoform A1 (*ALDH3A1*); this correlated with reduced levels of gene and promoter methylation, as well as the increased expression of *ALDH1A3*, and higher overall *ALDH* enzymatic activity. Moreover, 19 microRNAs were differentially expressed, overexpression was identified in members of the miR-29 family, and downregulation was exhibited in miR-708. NOY1-CR showed an increase in terms of migration, it formed 3D multicellular spheroids, and small micrometastasis occurred in the quail chorioallantoic membrane (CAM) model. In order to further confirm the tumorigenicity of NOY1-CR, these cells were injected subcutaneously into the flank of immunodeficient mice, which subsequently produced larger xenografts than the NOY1 parental cells. Importantly, the authors also showed that the cultured NOY1-CR cells were more sensitive to salinomycin and tunicamycin treatments compared with their parental cells. Moreover, combined treatment with the napabucasin augmented the toxicity of the cisplatin. Taken together, the data suggest that these drugs might be a potential treatment for refractory YST patients [103].

The NOY1 cell line was also used to investigate the interactions between peritoneal mesothelial cells and the maintenance of the stemness of human ovarian YST cells (SC-OYST) [104]. NOY1 cells were co-cultured with peritoneal mesothelial cells, and a high expression of CD133 was observed, in addition to a higher number of colonies of NOY1-CD133+, and a greater capacity for migration and invasion compared with NOY1-CD133-cells. When AMD3100 was added to co-culture systems in vitro, the colony formation, migration, and invasion of NOY1-CD133+ cells were inhibited. Moreover, when AMD3100 was used in vivo, it inhibited the tumorigenicity of the NOY1-CD133+ cells [104].

To continue evaluating the biological behavior of carcinomas in vitro, Iwasaki et al. established a novel YST cell line, TC587, taken from a 12-year-old girl with ovarian YST [105]. The cell line expressed AFP and SALL4, which are characteristic of YST. Moreover, next-generation sequencing was performed, and it revealed mutations in the *NRAS*, *KIT*, *KMT2C*, *RSF1*, and *TP53* genes. The newly established TC587 cell line may contribute to the individualization of YST treatments [105].

Although cell lines are the most commonly used model to study cancer, the in vivo model is the best model with which to reproduce the phenotypic properties of a tumor [106]; thus, it is essential to develop patient-derived xenograft (PDX) models that characterize ovarian GCTs in adult and pediatric patients. These models should focus on etiopathogenesis, histological type, metastases, and response to treatment. During the establishment of a PDX model, tumor samples from patients are freshly collected either during a diagnostic biopsy or during debulking surgery. Then, these tumor samples are minced and transplanted into immunocompromised mice (orthotopically or non-orthotopically) [107,108].

To date, no study has developed a PDX model for ovarian GCTs; however a PDX model showing the malignant transformation of mature cystic teratomas (MTMCT) in the ovary has been established, wherein tumor tissue was obtained from a 32-year-old patient with MTMCT [109].

Ovarian, primary peritoneal (PP), or fallopian tube tumors were collected at the time of surgery from patients (in the range of 22–91 years), and the samples were injected intraperitoneally into SCID mice [110]. Two hundred and forty-one models were injected, and 168 models were engrafted onto the mice. The grafted tumors exhibited a 74% engraftment rate, with microscopic faithfulness in terms of primary tumor characteristics and responses to carboplatin and paclitaxel in vivo; these results were associated with the corresponding patient’s clinical response [110].

Ricci et al. developed PDX models using tumor cells from clinical primary ovarian tumors and ascites fluid [111]. One hundred and thirty-eight patient samples were collected, and only 34 resulted in the formation of epithelial ovarian cancer xenografts, which meant that the original patients’ molecular and biologic features were able to be retained [111]. Heo et al. developed PDXs for epithelial ovarian cancer, the rate of successful PDX engraftment was 48.8% (22/45 cases), and the age of the patients when the cells were collected was 53.68 ± 10.18 years [112]. Moreover, the erlotinib treatment reduced tumor weight in PDXs of clear cell carcinoma with overexpression of epidermal growth factor receptor (EGFR) [112]. The ovarian yolk sac tumor (OYST) PDX model was developed from a patient (~1 year old) and treated with bleomycin, etoposide, and cisplatin (JEB). The chemotherapy regimens were consistent with the clinical outcomes of OYSTs, suggesting the PDX-OYST as a potential preclinical model [113].

The establishment of a PDX model of ovarian cancer may crucial for the development of novel therapies. Moreover, it may also clarify the carcinogenic mechanisms of ovarian cancer in the future. In addition, because there is no study that has an established PDX model for ovarian GCTs, either for adults or children, we believe that the determination of this model will allow a better understanding of the mechanisms involved in chemotherapy resistance, and it will provide a framework for the development of a precision medicine.

## 6. Molecular Implications of Treatment Resistance

This section is not mandatory but may be added if there are patents resulting from the work reported in this manuscript.

In 1978, cisplatin treatment was first approved by the Food and Drug Administration (FDA) for the treatment of testicular and bladder cancer. Currently, it is employed to treat a wide spectrum of solid tumors, including ovarian cancer [114,115,116]. Cisplatin frequently achieves an initial degree of outstanding success in terms of partial responses to treatment or disease stabilization; however, a clinically meaningful number of sensitive tumors eventually develop chemoresistance, which is habitually noticed in ovarian cancer patients [117,118,119,120].

Several molecular mechanisms have been associated with chemoresistance, including tumor suppressor genes, oncogenes, exosomes, DNA repair, cancer stemness, and epithelial–mesenchymal transition (EMT) [120,121] (Figure 3).

Nucleotide excision repair (NER) has been revealed as a key resistance system against CDDP in tumor cells. NER is a versatile DNA repair system and the NER pathway involves of several steps, including damage recognition, pre-incision complex assembly, dual incision, and repair synthesis and ligation. NER proteins can recognize cisplatin-induced DNA damage, remove them, and neutralize the cytotoxicity of CDDP, resulting in drug resistance [122].

Several studies have investigated the mechanisms involved in CDDP resistance in GCTs, and most of them have opted to examine the mechanisms involved in TGCTs. Consequently, little is known about such mechanisms with regard to ovarian GCTs. Given the common origin of TGCTs and OGCTs, it is likely that the mechanisms could work in a similar manner. The main mechanisms are described below.

The EMT is a biological program involving cells. It transiently converts epithelial cells into mesenchymal cells; indeed, during this process, epithelial cells gradually lose their characteristic phenotypes and they adopt heightened motility and invasiveness characteristics with a spindle-like morphology that lacks apical–basal polarity [86,123,124]. EMT induction can occur via transcription factors (EMT-TF) such as the SNAIL, SLUG, TWIST, and ZEB proteins, through the inactivation of cell junction proteins such as E-cadherin, claudins, and occluding proteins [125]. The role of EMT-inducing transcription factors (EMT-TFs) has been reported in several types of cancers, and our research has shown the importance of SNAIL EMT-TF in breast adenocarcinoma [126]. In addition, the ZEB1, SNAIL, and SLUG EMT-TFs were exhibited to confer resistance to oxaliplatin-based and cisplatin- based chemotherapy in breast, ovarian, colon, and pancreatic cancers [127,128]. Although the EMT process has been associated with several types of cancer, few studies have evaluated the role of EMT with regard to GCTs. A group of researchers from our research center have showed the key role that Brachyury EMT-TF plays in patients with TGCTs (in the range of 18–62 years) [129]. The first study that investigated the expression and potential clinical role of EMT-related factors in patients (in the range of 11–60 years) with malignant ovarian germ cell tumors (MOGCT) showed that EMT-related proteins are differentially expressed among MOGCT subtypes, thus suggesting differences between the biological characteristics associated with cell invasion and metastasis [130]. Although few studies have investigated EMT in GCTs, there are no studies exclusively evaluating EMT in pediatric patients; thus, evaluating the EMT process in GCT patients will allow a better understanding of the behavior of these tumors with regard to metastasis, recurrence, and drug resistance.

There is an attempt to elucidate the molecular mechanisms related to chemoresistance in yolk sac tumors as they are the second most common histological subtype of malignant ovarian GCTs; indeed, more than 50% of relapsed patients die of the disease. Zong et al. generated RNA-seq data for 12 YST samples (three sensitive primary tumors and nine relapsed tumors), and they showed that *OVOL2* (ovo-like zinc finger two) gene expression was significantly higher in relapsed tumors as compared with sensitive tumors. Moreover, the overexpression of *OVOL2* was associated with cisplatin resistance in the cancer cell lines of several lineages, including ovarian cancer cell lines. These results suggest that the high expression of *OVOL2* may be correlated with the mechanism of chemoresistance in YSTs [65].

One of the main mechanisms of post-target resistance involves the inactivation of *TP53* [131], which is reported to happen in almost half of all human tumors [132]. GCTs frequently exhibit a high expression of wild-type P53 in the cytoplasm and nuclei of its cells, although, such expressions are inconsistent [133,134]. Only a subset of refractory TGCTs exhibiting CDDP resistance have been directly associated with *TP53* mutations [135], which suggests that mutations in the regulators of the P53 pathway play a significant role in CDDP resistance with regard to TGCTs. Based on the TGCT data, ovarian cancer patients with the wild-type *TP53* mutation are more likely to benefit from CDDP-based chemotherapy compared with patients who have *TP53* mutations [136,137].

Furthermore, a study evaluated the role of P53 and MDM2 in terms of the sensitivity and resistance of TGCTs to cisplatin-based chemotherapy in adolescents and adults [138]. The analysis showed only one silent P53 mutation in one of the responding patients and the amplification of MDM2 was found in one out of twelve embryonal carcinomas. The result concerning the presence of the wild-type P53 mutation in TGCTs is in accordance with previous studies; however, it has been suggested that a high number of P53 mutations does not directly correlate with the sensitivity of these tumors to treatment, and P53 inactivation is not directly involved in the mechanisms that concern cisplatin resistance [138]. Thus, more studies are required to explain the association between the P53 mutation and CDDP chemosensitivity in GCTs.

## 7. Conclusions

A widespread variety of chemotherapy regimens have been used for OGCTs, and differences and similarities are found with regard to the management of OGCTs in pediatric and adult patients; however, further studies are needed to determine which drugs can be used in order to avoid overtreatment over a certain period of time.

Discrepancies between molecular analyses, in terms of the publications that focus on OGCTs in both adult and child patients, are common; however, the reasons for these discrepancies may be largely explained by the limited number of samples, or the fact that several different techniques have been used.

In both sexes, TGCTs and OGCTs develop via several of the same pathogenetic mechanisms; moreover, they also share similar molecular characteristics. As testicular GCTs recur more frequently and, thus, have been studied more extensively, new insights and data that have been obtained in clinical trials can also be applied to OGCTs; this will increase the number of therapeutic options when attempting to treat this rare disease. The identification of potential molecular alterations may provide a novel area for further research, particularly in terms of understanding the pathogenesis, tumorigenesis, diagnostic markers, and genetic peculiarity of this rare tumor type in the ovary.

## Figures and Tables

**Figure 1 cancers-15-02990-f001:**
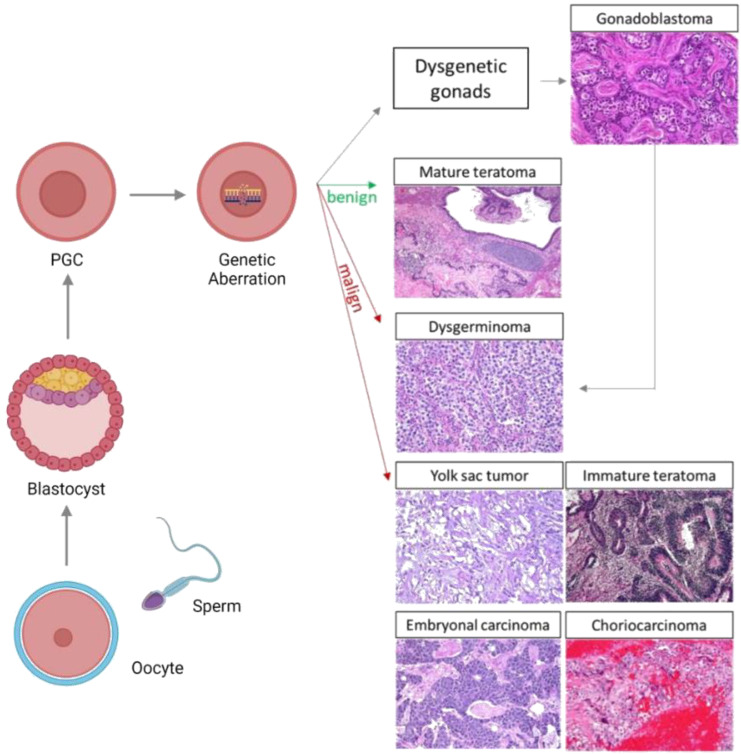
Patients with disordered female gonadal development, caused by various genomic aberrations, can present gonadal dysgenesis with a high risk of developing malignant OGCTs; such tumors often develop from gonadoblastomas. Genomic abnormalities in PGCs can also develop into benign tumors as mature teratomas, and malign tumors including dysgerminomas, yolk sac tumors, immature teratomas, embryonal carcinoma, and choriocarcinoma may also develop. The green arrows represent benign tumors and the red arrows represent malign tumors. PGC: Primordial germ cell.

**Figure 3 cancers-15-02990-f003:**
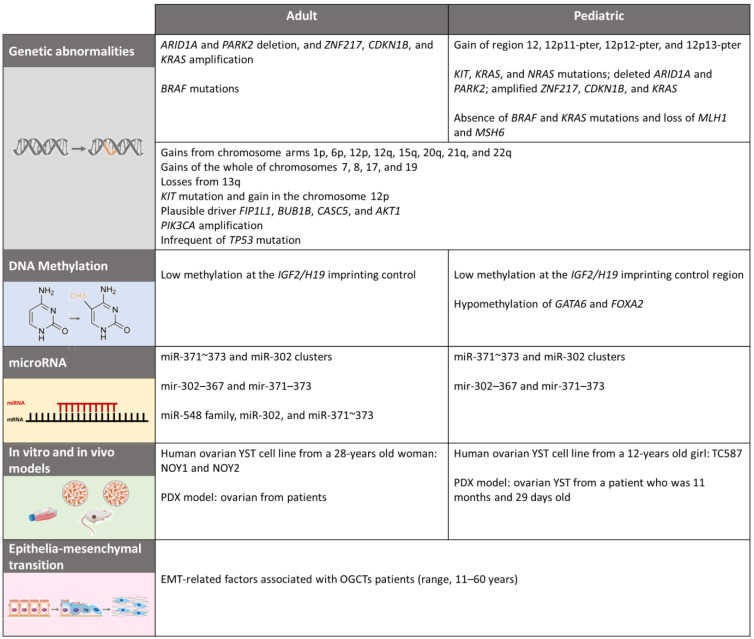
Comparison of molecular differences between adult and pediatric patients with OGCTs.

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
