# Peer review of "Molecular Biology of Pediatric and Adult Ovarian Germ Cell Tumors: A Review"

_cancers, 2023, doi:10.3390/cancers15112990_

Round 1
Reviewer 1 Report
The manuscript of Pinto and colleagues aims to provide a description of pediatric and adult ovarian germ cell tumors, focusing on etiopathogenesis, development, therapeutic approaches, molecular biology, in vitro and in vivo models and molecular implications of treatment resistance. It follows a previous review by the same group on Pediatric and Adult Male Germ Cell Tumors (Pinto et al., Cancers 2021). Besides, the research group show a great expertise in germ cell tumor field. The current review is well structured, and all information are accurately reported. In my opinion, it is suitable for publication in this form following minor revision.
Minor comments:
-According to Cancer Statistics 2019, Germ cell & gonadal tumors (no exclusively ovarian germ cell tumors) account for 11% of cancer diagnosis in 15-19 years old group (lines 35-36). Please update with cancer statistics 2023.
-In lines 170-172, please add percentage of incidence of OGCTs other than Dysgerminoma and immature teratoma (i.e. YSTs and mixed GCTs).
-To improve understanding of concepts, please insert figure captions into the text.
-Lines 718-719 are out of context, please remove these sentences.
-Authors should shorten the general statements and related references (e.g. lines 720-729; lines 548-555). In addition, Authors should remove statements which are not the main objective of the review (e.g. male germ cell tumors or high grade serous ovarian cancer).
Author Response
Reviewer 1
- According to Cancer Statistics 2019, Germ cell & gonadal tumors (no exclusively ovarian germ cell tumors) account for 11% of cancer diagnosis in 15-19 years old group (lines 35-36). Please update with cancer statistics 2023.
Response: Thank you for pointing that out. We have updated the data according to cancer statistics 2023.
- In lines 170-172, please add percentage of incidence of OGCTs other than Dysgerminoma and immature teratoma (i.e. YSTs and mixed GCTs).
Response: We appreciate the suggestion of the reviewer to add percentage of incidence of OGCTs other than dysgerminoma and immature teratoma. Therefore, and the sentence has been restructured: “Dysgerminoma and immature teratoma are the most common types of OGCT, comprising 65-70% of all OGCTs, followed by YSTs (14.5%), and finally, mixed GCTs (5.3%)”.
- To improve understanding of concepts, please insert figure captions into the text.
Response: We totally agree with the reviewer and we have inserted the figure captions into the text.
- Lines 718-719 are out of context, please remove these sentences.
Response: We have removed the sentence: “In preclinical tests, the paclitaxel-carboplatin combination chemotherapy decreased tumor weight in PDXs compared with the control treatment”.
- Authors should shorten the general statements and related references (e.g. lines 720-729; lines 548-555). In addition, Authors should remove statements which are not the main objective of the review (e.g. male germ cell tumors or high grade serous ovarian cancer).
Response: Thank you for pointing that out. We have shortened the following sentence: “The ovarian yolk sac tumor (OYST) PDX model was developed from a patient (~1 year old) and treated with bleomycin, etoposide, and cisplatin (JEB). The chemotherapy regimens were consistent with the clinical outcomes of OYSTs, suggesting the PDX-OYST as a potential preclinical model”.
The following sentences have been removed:
1) ”The role of DNA methylation with regard to cisplatin resistance was also investigated using the seminomatous cell line, TCam-2. TCam-2 cells treated with 5-azacytidine showed a significant degree of hypomethylation in 54 CpG sites (23.5%), and hypermethylation in 176 loci (76.5%). The KLF11 gene, exhibited a highly demethylated CpG site, and the CFLAR and ERBB2 showed hypermethylated CpG sites [79]. Further experiments are necessary to explore the potential of 5-azacytidine in the treatment of GCTs that are resistant to cisplatin”.
2) “Similarly, the PDX model, obtained from high-grade serous ovarian cancer (HG-SOC), was generated, and the success rate of xenografting was 83% [111]. In addition, all ten HG-SOC PDXs analyzed contained mutations in TP53; indeed, two were mutated for BRCA1, three for BRCA2, and in two, BRCA1 was methylated. The in vivo cisplatin response was largely consistent with the patient’s outcome [111]. These tumorgrafts are valuable to better understanding of the mechanisms involved in chemotherapy resistance”.
Reviewer 2 Report
This review deals with hot topics in medicine and molecular biology: understanding of tumorigenesis molecular basis and women’s fertility preservation. Authors collected and systematized a large amount of published data that could be of interest for the scientific and medical communities. Text is easy to read and understand.
Manuscript can be accepted for publication after a following minor revision:
· It is known that nucleotide excision repair plays a role in the platinum-based therapy resistance, so a couple of sentences about this DNA repair pathway involvement will benefit the review.
Author Response
Reviewer 2
- It is known that nucleotide excision repair plays a role in the platinum-based therapy resistance, so a couple of sentences about this DNA repair pathway involvement will benefit the review.
Response: We appreciate the suggestion of the reviewer to include a couple of sentences about nucleotide excision repair (NER). A paragraph has been added in the topic “6. Molecular implications of treatment resistance”